# A Measurement-Data-Driven Control Approach towards Variance Reduction of Micromachined Resonant Accelerometer under Multi Unknown Disturbances

**DOI:** 10.3390/mi10050294

**Published:** 2019-04-30

**Authors:** Qiang Shen, Dengfeng Yang, Jie Zhou, Yixuan Wu, Yinan Zhang, Weizheng Yuan

**Affiliations:** 1Research & Development Institute of Northwestern Polytechnical University in Shenzhen, Shenzhen 710000, China; 2MOE Key Laboratory of Micro and Nano Systems for Aerospace, Northwestern Polytechnical University, Xi’an 710072, China; fdy1013@163.com (D.Y.); wuyixuan@mail.nwpu.edu.cn (Y.W.); zyn@mail.nwpu.edu.cn (Y.Z.); 3Department of mechatronic engineering, Northeast Forestry University, Harbin 150040, China; 4School of electronic information engineering, Xi’an Technological University, Xi’an 710021, China; zhoujie@xatu.edu.cn

**Keywords:** microelectromechanical system (MEMS) accelerometer, variance characteristic, unknown disturbance control, data-driven model

## Abstract

This paper first presents an adaptive expectation-maximization (AEM) control algorithm based on a measurement-data-driven model to reduce the variance of microelectromechanical system (MEMS) accelerometer sensor under multi disturbances. Significantly different characteristics of the disturbances, consisting of drastic-magnitude, short-duration vibration in the external environment, and slowly-varying, long-duration fluctuation inside the sensor are first constructed together with the measurement model of the accelerometer. Next, through establishing a data-driven model based on a historical small measurement sample, the window length of filter of the presented algorithm is adaptively chosen to estimate the sensor state and identify these disturbances simultaneously. Simulation results of the proposed AEM algorithm based on experimental test are compared with the Kalman filter (KF), least mean square (LMS), and regular EM (REM) methods. Variances of the estimated equivalent input under static condition are 0.212 mV, 0.149 mV, 0.015 mV, and 0.004 mV by the KF, LMS, REM, and AEM, respectively. Under dynamic conditions, the corresponding variances are 35.5 mV, 2.07 mV, 2.0 mV, and 1.45 mV, respectively. The variances under static condition based on the proposed method are reduced to 1.9%, 2.8%, and 27.3%, compared with the KF, LMS, and REM methods, respectively. The corresponding variances under dynamic condition are reduced to 4.1%, 70.1%, and 72.5%, respectively. The effectiveness of the proposed method is verified to reduce the variance of the MEMS resonant accelerometer sensor.

## 1. Introduction

Microelectromechanical system (MEMS) technologies have successfully enabled the miniaturization and cost reduction of many kinds of sensors. These MEMS sensors have been widely used in many fields [1,2,3], including artificial intelligence (AI), internet of things (IoT), and industry 4.0. Especially, the MEMS resonant accelerometer, as a kind of functional but important device, has recently focused lots of research interest due to its potentially high accuracy and built-in digital characteristic [4,5]. However, currently, the comparatively low performance of this type of accelerometer severely limits the application range, which suffers from lots of factors, such as scale-factor errors, bias drift, and additively harmful pressure and shock effects [6,7]. Among these factors, the comparatively large bias drift of the resonant accelerometer is one major reason to deteriorate its variance. Especially during a sharp vibration and shock occurrence, the bias of the accelerometer sensor will fluctuate dramatically, which leads to a terrible performance deterioration due to its sophisticated, but fragile mechanical structure [8]. Even worse, it is not easy to suppress these disturbances by common compensation methods, such as mechanical topology design [9,10,11], closed-loop circuit [5,12,13], and package technology [14,15,16] due to unknown and uncertain characteristic of these harmful inputs.

Signal post-processing technology is an effective way to estimate the unknown inputs in some fields, such as sensor modelling [17,18,19], object tracking [20,21,22], and coupled audio-visual analysis [23]. The bias model of MEMS sensors with unknown inputs (UIs) caused by inaccurate coefficient values, parameter variations, and uncertain disturbances can be estimated by establishing adaptive or robust controllers based on this post-processing technology. For example, a reduced-order disturbance decoupled observer is designed based on a parametric approach to jointly estimate the sensing system state and input with UI [24,25]. Further, a parameter-dependent robust mixed full- and reduced-order controllers for discrete-time sensing system with UI are obtained via the linear matrix inequality optimization [26]. UI of sensors are also modelled as a randomly switching parameter obeying a known Markov chain in multiple model estimation [27], where the corresponding controller contains variable-structure MME, interacting MME, and pseudo-Bayesian estimation. Another approach based on minimum upper bound filter (MUBF) is used to estimate a statistically-constrained UI by establishing the minimized the upper bounds of covariance matrices of the state prediction error and residual error [28].

For the MEMS resonant accelerometer sensor, the corresponding variance is usually affected by two types of inputs. The one, is vibration and shock with a characteristic of drastic amplitude and short duration sourcing from external environment. The other one, is a slowly varying fluctuation sourcing from the accelerometer’s interior, such as thermal, pressure drift, and stress release of mechanical components. These two inputs can be commonly regarded as unknown parameters with stochastic and uncertain characteristics.

Currently, the data-driven-based signal processing approach is becoming a potentially interesting research to estimate the uncertain input of sensors [29,30,31,32,33]. The statistical models of sensors are fitted upon the available historical and condition data. In this work, the adaptive expectation-maximization (EM) control algorithm based on the data-driven model is first proposed to reduce the variance of the MEMS resonant accelerometer sensor under multi unknown disturbances. Two types of the external and internal disturbances are first considered simultaneously. Through constructing a data-driven model based on historical small sample data, a smoother window-length is adaptively chosen. The external and internal disturbances with significantly different characteristics are identified and the bias of the accelerometer is finally estimated.

This paper is organized as follows: In Section 2, a model of the MEMS resonant accelerometer sensor under multi unknown disturbances is established. In Section 3, the joint estimation and identification algorithm based on the measurement-data-driven AEM are estimated and simulated. In Section 4, the experimental test of the proposed AEM method is compared and verified. Finally, the conclusions are given in Section 5.

## 2. Modelling of the Accelerometer Output Under Unknown Input

A typical MEMS resonant accelerometer sensor consists of a mechanical element and electrical element, as shown in Figure 1. *y* represents the output of MEMS accelerometer. When unknown inputs including external disturbance and internal disturbance occur, discrete-time model of output *y_k_* at time point *t_k_* can be first expressed as follows:(1)$$y_{k}=s_{k}+u_{ex,k}+u_{in,k}+w_{k}$$
where *s_k_* represents the true equivalent acceleration input of the MEMS accelerometer sensor. *u*_ex_*_,k_* represents the equivalent input induced by an external unknown disturbance at time point *t_k_* with the characteristic of drastic amplitude and short duration, such as shock and vibration. *u*_in_*_,k_* represents the equivalent input caused by an internal unknown disturbance at time point *t_k_*, such as stress release of the mechanical element and the thermal drift of electrical element components. *w_k_* represents zero-mean white Gaussian noise. Here, let *u_k_* = *u*_ex_*_,k_* + *u*_in_*_,k_*.

A state-space model for the MEMS accelerometer sensor is first constructed. In most applications of MEMS sensors, the maneuverability characteristics of aircrafts can be regarded as constraining in a certain frequency bandwidth and magnitude. Additionally, the true input signal *s_k_*_+1_ of the sensors for the subsequent time point *t_k_*_+1_ usually can be considered as relating to *s_k_* for the former time point *t_k_*. The above two characteristics can be suitably described by a first-order Markov process with a process time constant set according to the system’s bandwidth and the process noise related to the limit of the magnitude of the motion [34,35,36]. Hence, according to Equation (1), the dynamical and measurement models of the true input signal for MEMS accelerometer can be constructed, respectively.(2)$$x_{k+1}=\Phi_{k}x_{k}+v_{k}$$
(3)$$z_{k+1}=H_{k+1}x_{k+1}+\Gamma_{k+1}u_{ex,k+1}+\Psi_{k+1}u_{in,k+1}+w_{k+1}$$
where *k* is the sampling time, *x_k_* = *s_k_* represents the state of accelerometer, *z_k_*_+1_ represents the measurement of sensor, *u*_ex_*_,k_*_+1_ and *u*_in_*_,k_*_+1_ are unknown external and internal disturbance, respectively. Similarly with Equation (1), let *u_k_*_+1_ = *Γ_k_*_+1_
*u*_ex_*_,k_*_+1_ + *ψ_k_*_+1_
*u*_in_*_,k_*_+1_. Defining $z_{k-v+1}^{k}$ = {*z_k−v_*_+1_, …, *z_k_*}, measurement noise *w_k_*_+1_ with *E*[*w_k_*] = 0 and *E*[*w_k_ w^T^_k_*_+*τ*_] = ***R**δ(τ)* is characterized by the white noise with covariance matrix ***R*** = *σ_R_*^2^
***I*_1×1_** that can be obtained by calculating the final output data of the single sensor based on the Allan variance method. *δ(τ)* is the Dirac delta function. Process noise ***v_k_*** with *E*[*v_k_ v^T^_k_*_+*τ*_] = ***Q**δ(τ)* is described by a covariance matrix ***Q** = σ_Q_*^2^
***I*_1×1_** that can be set according to the dynamic characteristic of the true input signal. *σ_Q_*^2^ and *σ_R_*^2^ are the corresponding variance parameters, respectively. Coefficient scalar *Φ_k_* = −1/*τ_s_*, where *τ_s_* is process time constant based on first-order Markov process. *H_k_*_+1_, *Γ_k_*_+1_ and *ψ_k_*_+1_ are all one-dimensional identity matrixes. Here, symbol **Θ** represents vector {*u*_ex_, *u*_in_, *σ_Q_*^2^, *σ_R_*^2^}.

## 3. Joint Estimation and Identification Strategy Based on the Data-Driven Iterative Optimization

To perform the identification of the unknown input *u*_ex_, *u*_in_ and the estimation of the true acceleration input signal *s* simultaneously, it is necessary to use an iterative optimization strategy to achieve joint estimation and identification because unknown input identification mistake deteriorates the state estimate while the state estimate error increases the identification risk. The EM algorithm is an effective approach to implement this iterative optimization. Nevertheless, due to the significant differences in the characteristics of magnitude and duration between the external and internal disturbances, it is not suitable for the EM algorithm to keep the window length of its filter at a constant. Hence, the AEM algorithm driven by measurement data of MEMS accelerometer sensor under complex disturbances is first presented to achieve iterative optimization, and the following derivation follows the idea of the proposed data-driven-based AEM strategy.

The function flow of the proposed method is shown in Figure 2. The proposed AEM method mainly consists of a likelihood function estimation, variable window-length smoother selection, state estimation in E step, parameter identification in M step and iteration decision. The detailed proposed method of the MEMS accelerometer sensor is given as follow.

The likelihood function of measurement *z_k_* of the MEMS accelerometer is first given to establish the corresponding maximum expectation. Then, in E-step and M-step stages, the maximum expectation of the log likelihood function is performed by these two iteration steps. The unknown external and internal disturbances are then identified based on their difference on window lengths of filters, which are driven by slope of acquired measurement data over a certain period of time. Finally, equivalent acceleration input of the sensor is estimated.

### 3.1. Likelihood Function of the Accelerometer Sensor Measurement

For the maximum-likelihood estimation of the MEMS accelerometer sensor, the complete-data log-likelihood function (CLLF) is defined to be the logarithm of the probability density function of output observations parameterized with **Θ**. First, the complete-data LLF for the parameter **Θ** can be written as follows:(4)$$\begin{array}{cc} L_{k-v+1}^{k} & =\log p\left(x_{k-v+1},\ldots ,x_{k},z_{k-v+1},\ldots ,z_{k}|\Theta_{k-v+1}^{k},z_{1}^{k-v}\right)\\ & =\log p\left(x_{k-v}|z_{1}^{k-v}\right)+\sum_{i=k-v+1}^{k}\log p\left(x_{i}|x_{i-1},\Theta_{k-v+1}^{k}\right)\\ & \;\;+\sum_{i=k-L+1}^{k}\log p\left(z_{i}|x_{i},\Theta_{k-v+1}^{k}\right) \end{array}$$
where *v* is the window length of the filter, *p* is the probability density function of the output measurement of the MEMS accelerometer. The function *p* of the above expression is assumed to obey Gaussian distribution and is expressed as follows:(5)$$\left\{ \begin{array}{l} p\left(x_{k-v}|\Theta_{1}^{k-v}\right)\sim N\left({\hat{x}}_{k-v},Q_{k-v}\right)\\ p\left(x_{i}|x_{i-1},\Theta_{k-v+1}^{k}\right)\sim N\left(\Phi_{i-1}x_{i-1},Q_{i-1}\right)\\ p\left(z_{i}|x_{i},\Theta_{k-v+1}^{k}\right)\sim N\left(H_{i}x_{i}+u_{i},R_{i}\right) \end{array}\right.$$

Further, the log-likelihood function $L_{k-v+1}^{k}$ is given in detail as follows:(6)$$L_{k-v+1}^{k}=L_{0,k-v+1}^{k}+L_{1,k-v+1}^{k}+L_{2,k-v+1}^{k}+L_{3,k-v+1}^{k}$$
with the following:(7)$$L_{0,k-v+1}^{k}=-\frac{n+\left(L+2\right)\left(d_{1}+d_{2}\right)}{2}\log\left(2\pi \right)-\frac{1}{2}\sum_{i=k-v+1}^{k}\log\left(\left|Q_{i}\right|+\left|R_{i}\right|\right)$$
(8)$$L_{1,k-v+1}^{k}=-\frac{1}{2}{\left(x_{k-v}-{\hat{x}}_{k-v}\right)}^{T}Q_{k-v}^{-1}\left(x_{k-v}-{\hat{x}}_{k-v}\right)$$
(9)$$L_{2,k-v+1}^{k}=-\frac{1}{2}\sum_{i=k-v+1}^{k}{\left(x_{i}-\Phi_{i-1}x_{i-1}\right)}^{T}Q_{i-1}^{-1}\left(x_{i}-\Phi_{i-1}x_{i-1}\right)$$
(10)$$L_{3,k-v+1}^{k}=-\frac{1}{2}\sum_{i=k-v+1}^{k}{\left(z_{i}-H_{i}x_{i}-u_{i}\right)}^{T}R_{i}^{-1}\left(z_{i}-H_{i}x_{i}-u_{i}\right)$$
where the dimensions *d*_1_ and *d*_2_ of the state *x* and measurement *z* are equal to 1, respectively. To estimate the state and identify the parameter of the MEMS accelerometer, maximum expectation of the log likelihood function is performed by the proposed AEM algorithm below.

### 3.2. Joint Estimation and Identification based on Iterative Optimization

The AEM algorithm to identify the disturbances and estimate the above equivalent input of MEMS accelerometer is first given as follows:E step:(11)$$Q\left(\Theta_{k-v+1}^{k}|\Theta_{k-v+1}^{k|r}\right)=E_{x_{k-v+1}^{k|r}|z_{k-v+1}^{k},\Theta_{k-v+1}^{k|r}}\left(L_{k-v+1}^{k}\right)$$
where r is the number of iterations, *Q*(*r*) represents $Q\left(\Theta_{k-v+1}^{k}|\Theta_{k-v+1}^{k|r}\right)$M step:(12)$$u_{k-v+1}^{k|r+1}{=}_{u_{k-v+1}^{k}}^{\arg\max}Q\left(\Theta_{k-v+1}^{k}|\Theta_{k-v+1}^{k|r}\right)$$

The corresponding maximum expectation value is represented as $Q_{\max}^{r}$ because these partial derivations should be zero at the optimal point of state *x* estimation and disturbance *u* identification; therefore:(13)$$\frac{\partial Q\left(\Theta_{k-v+1}^{k}|\Theta_{k-v+1}^{k|r}\right)}{\partial u_{k-v+1}^{k}}=\sum_{i=k-v+1}^{k}\frac{\partial E\left[{\left(z_{i}-H_{i}x_{i}-u_{i}\right)}^{T}R_{i}^{-1}\left(z_{i}-H_{i}x_{i}-u_{i}\right)\right]}{\partial u_{i}}$$

Here, *u_i_* is equal to each other within the window length of *v* as follows:(14)$$u_{i_{k}}^{r}=u_{i_{k-1}}^{r}=\ldots =u_{i_{k-v+1}}^{r}$$

Therefore, we obtain the following:(15)$$u_{i_{k-v+1}}^{k|r+1}={\left[{\left(\sum_{i=k-v+1}^{k}R_{i}^{-1}\right)}^{T}\left(\sum_{i=k-v+1}^{k}R_{i}^{-1}\right)\right]}^{-1}\times {\left(\sum_{i=k-v+1}^{k}R_{i}^{-1}\right)}^{T}\left(\sum_{i=k-v+1}^{k}R_{i}^{-1}\left(z_{i}-H_{i}{\hat{x}}_{i}\right)\right)$$

Then, the (r + 1)^th^-cycle $\Theta_{k-v+1}^{k|r+1}$ is iterated by replacing $u_{i_{k-v+1}}^{k|r}$ in $\Theta_{k-v+1}^{k|r}$ with the above $u_{i_{k-v+1}}^{k|r+1}$.

Iteration will stop when any one of the below two cases occurs. The one is the number of iterations reaches the upper limit $r_{\max}$ set by the simulation. The other one is the increment of the likelihood function between two consecutive iterations is less than the limit value δ, which means the following:(16)$$\Delta Q=\frac{Q\left(\Theta_{k-v+1}^{k}|\Theta_{k-v+1}^{k|r+1}\right)-Q\left(\Theta_{k-v+1}^{k}|\Theta_{k-v+1}^{k|r}\right)}{Q\left(\Theta_{k-v+1}^{k}|\Theta_{k-v+1}^{k|r+1}\right)}<\delta$$
where the range of δ is usually set as (0, 1), *Q*(*r* + 1) represents $Q\left(\Theta_{k-v+1}^{k}|\Theta_{k-v+1}^{k|r+1}\right)$. To take the balance between output accuracy with the running speed of the algorithm, here δ is 0.01.

### 3.3. Adaptive Selection on Window Length of Filter

Due to the different characteristics of *u*_ex_ and *u*_in_, the window-length of the filter is adaptively chosen. The larger smoothing window length of the filter is suitable to suppress the long-time slowly varying disturbance of the sensors effectively, such as *u*_in_, while the small window length is more effective to reduce short-time drastic fluctuation, such as *u*_ex_. In detail, slope variation criterion driven by historical small sample data is constructed to determine the window length of the filter in order to suppress these two kinds of disturbances simultaneously. Here, the window length model of the filter is formulated as follows:(17)$$v=\{\begin{array}{cc} 15 & if\;\left|\Delta k_{s}\right|\leq \frac{\sqrt{3}}{3}\\ 10 & if\;\frac{\sqrt{3}}{3}<\left|\Delta k_{s}\right|\leq \sqrt{3}\\ 5 & if\;\left|\Delta k_{s}\right|>\frac{\sqrt{3}}{3} \end{array}$$
where Δ*k_s_* = max *k_s_*{*z_i_*’, *z_j_*’} − min *k_s_*{*z_i_*’, *z_j_*’} represents slope difference between the maximum slope value and the minimum value, *k_s_*{*z_i_*’, *z_j_*’} represents the slope value between any two points on the curve fitted based on historical sample measurement data. *z_i_*’ and *z_j_*’ are the new fitted sample points corresponding to historical sample points *z_i_* and *z_j_*, respectively. The recursive computation process of the derived AEM algorithm is summarized in Table 1.

## 4. Experimental Test and Comparison

The test system of the MEMS sensor is mainly composed of a resonant accelerometer device [37], control circuit, data acquisition unit using Agilent 34410 with sampling frequency of 15 Hz to collect the output of sensors, and the PC that is connected to 34410 by USB-port communication interface and online process the measurement data by the proposed method. Figure 3 shows the overall test system of the MEMS sensor.

### 4.1. Static Performance Test

The MEMS resonant accelerometer sensor is performed to respond to the static output under stochastic disturbances with two types of characteristics of slowly varying fluctuation and drastic-amplitude and short-duration vibration. The results of the proposed method are compared with other vibration suppression algorithms, such as the LMS algorithm, the KF algorithm and the REM algorithm. The variance parameters are initially set as *σ_Q_* = 0.93 and *σ_R_* = 0.99.

First, some groups of stochastic signals, including a slowly varying fluctuation and drastic-amplitude, short-duration vibration are generated randomly. Figure 4 shows the estimated equivalent input by different algorithms. First, the detailed characteristics of the three groups of the disturbances are listed in Table 2. The equivalent weight of movable platform of the vibration table is 2 kg and the corresponding impact accelerometer is 490 m/s^2^.

The corresponding results are listed in Table 3. The effectiveness of the proposed algorithm is proven by the variance of the equivalent input of 0.004 mV, compared with variances of 0.212 mV, 0.149 mV, and 0.015 mV estimated by KF, LMS, and REM algorithms, respectively. The corresponding variance by the proposed method is reduced to 1.9%, 2.8%, and 27.3%, respectively.

### 4.2. Dynamic Performance Test

The dynamic characteristic of the MEMS resonant accelerometer under the disturbances is also compared with other fluctuation filtering algorithms to verify the effectiveness of the proposed method. The input acceleration has the amplitude and frequency of 2 mV and 0.2 Hz with a peak-to-peak value 0.11 mV of white Gaussian noise, respectively.

Further, the disturbances of the impulse and squared waves are applied to the resonant accelerometer randomly. The maximum amplitudes of these impulse signals are within the equivalent input voltage of ±60 mV. The detailed characteristics of the three groups of the squared wave and impulse disturbances are listed in Table 4. The equivalent input is estimated by different algorithms, as shown in Figure 5.

The variance of the equivalent input estimation is listed in Table 5. The variance of the proposed algorithm is much smaller than the variances of other methods, which is decreased to 4.1%, 70.1%, and 72.5%, respectively. The effectiveness of the proposed method is proven.

## 5. Conclusions

The AEM control algorithm driven by measurement data is first established to reduce the variance of the MEMS resonant accelerometer sensor under multi unknown disturbances. The corresponding simulations based on experimental test are analyzed by comparing the proposed method with the KF, the LMS, and the REM methods. The variances of the equivalent input estimation of MEMS accelerometer with multi disturbances under static and dynamic conditions are 0.004 mV and 1.45 mV, respectively. The variances under static condition based on the proposed method are reduced to 1.9%, 2.8%, and 27.3%, compared with KF, LMS, and REM method, respectively. The corresponding variances under dynamic condition are reduced to 4.1%, 70.1%, and 72.5%, respectively. The effectiveness of the proposed method is verified to reduce the variance of the MEMS resonant accelerometer sensor.

## Figures and Tables

**Figure 1 micromachines-10-00294-f001:**
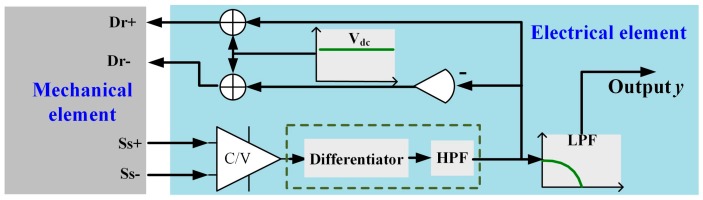
Schematic of a typical microelectromechanical system (MEMS) resonant accelerometer sensor.

**Figure 2 micromachines-10-00294-f002:**
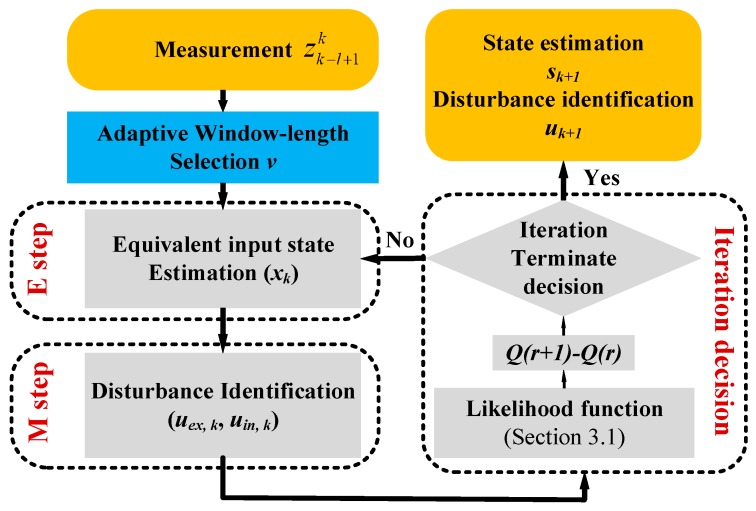
Function flow of the proposed adaptive expectation-maximization (AEM) method.

**Figure 3 micromachines-10-00294-f003:**
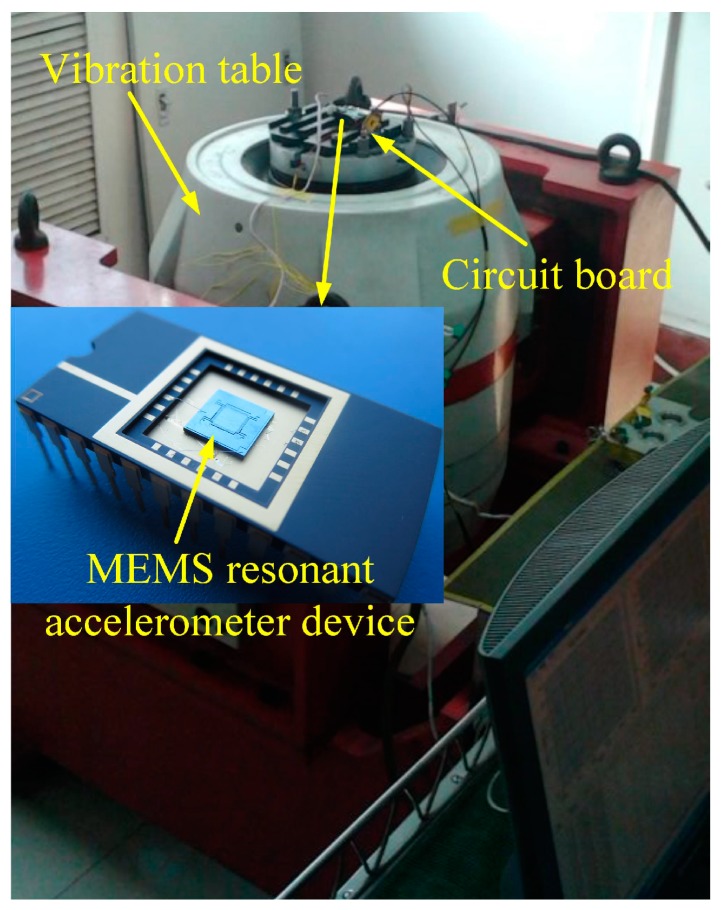
Test system of the MEMS resonant accelerometer sensor.

**Figure 4 micromachines-10-00294-f004:**
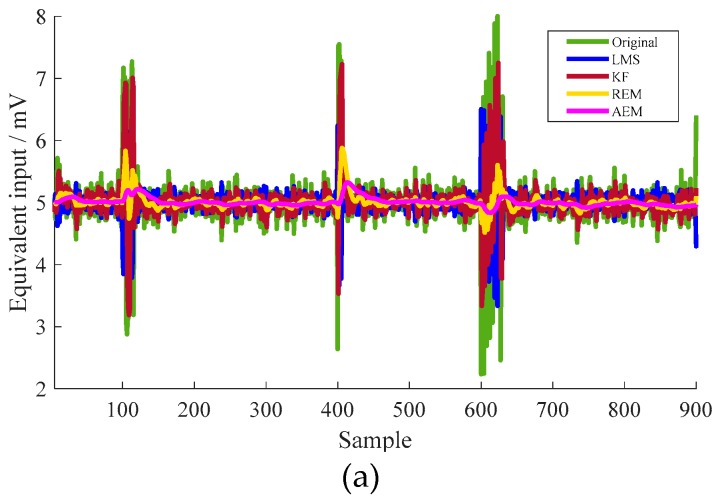
(**a**) Equivalent input estimation of the MEMS resonant accelerometer under some disturbances, (**b**) detailed view of the drastic disturbance at approximately 600th sample point, (**c**) detailed view of slowly-varying disturbance in the range between 200th and 350th sample points, (**d**) adaptive window-length variance of the proposed filter under static condition.

**Figure 5 micromachines-10-00294-f005:**
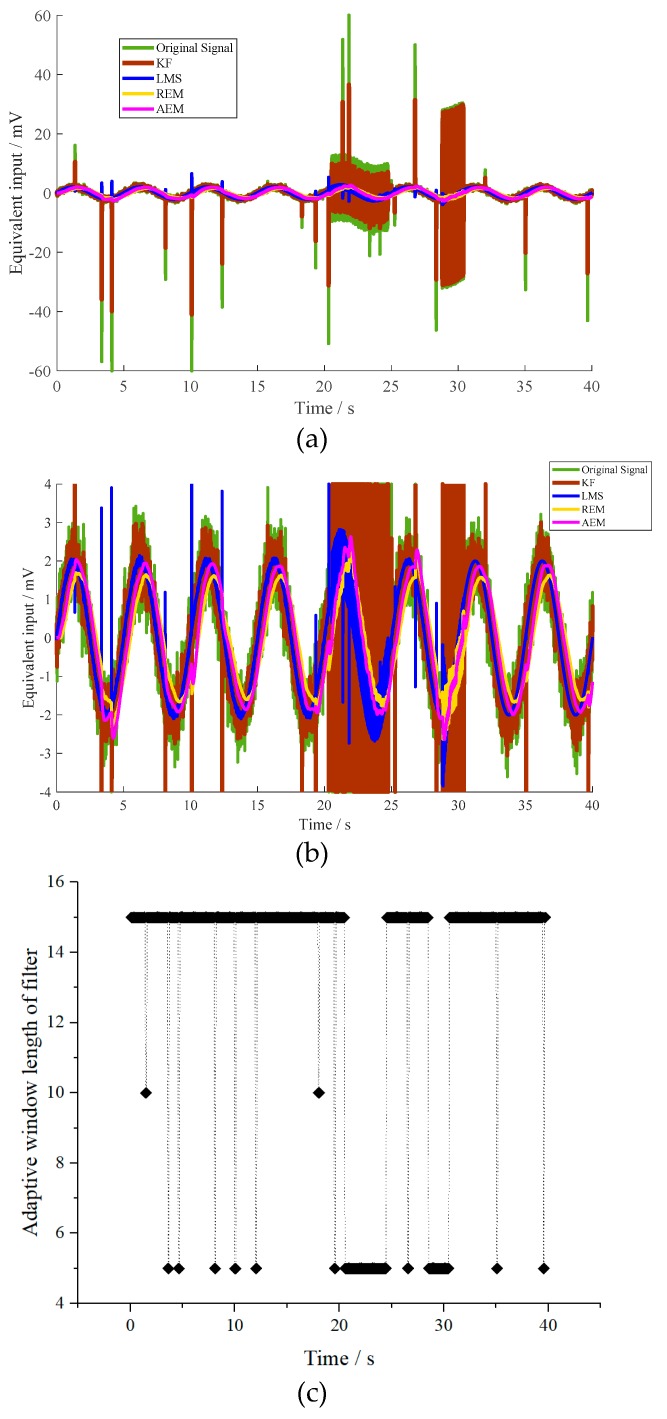
(**a**) Equivalent input estimation of the MEMS resonant accelerometer under dynamic condition with multi disturbances, (**b**) detail view of estimation results from −4 mV to 4 mV, (**c**) adaptive window-length variance of the filter under dynamic condition.

**Table 1 micromachines-10-00294-t001:** Recursive computation process of the proposed method.

**Step**	**Description**
**1). Preparation**	Derivative likelihood functions $L_{k-v+1}^{k}$
**2). Start**	Set window length *v* of the filter
**3). Obtain Data**	Obtain measurement sequence $z_{k-l},\ldots ,z_{k}$, as shown in Figure 1.
**Iterate the joint estimation and identification algorithm: for *r* = 1, 2, …**	
**4). E-Step**	Use KF to estimate ${\hat{x}}_{k-v+1}^{k|r}$, then calculate the expectation value $E_{x_{k-v+1}^{k|r}|z_{k-v+1}^{k},\Theta_{k-v+1}^{k|r}}\left(L_{k-v+1}^{k}\right)$ of the log likelihood function $L_{k-v+1}^{k}$
**5). M-Step**	Identify the parameters $u_{k-v+1}^{k|r^{1}+1}$ to ensure the above expectation value to reach the maximum value $Q_{\max}^{r}$
**6). Termination**	Calculate ΔQ in Equation (20), If ΔQ<δ or $r>r_{\max}$, then terminate the current number of iterations as r. Else, update $\Theta_{k-v+1}^{k|r}$ as $\Theta_{k-v+1}^{k|r+1}$ by replacing $u_{k-v+1}^{k|r}$ with the above $u_{k-v+1}^{k|r+1}$, and go to step 4.
**7). Result**	Obtain optimal ${\hat{s}}_{k-v+1}^{k}$

**Table 2 micromachines-10-00294-t002:** Characteristic of multi disturbances in static test.

Group	Shock Displacement (mm)	Number of Shock	Cycles Time (s)
**1**	0.18	2	1
**2**	0.2	1	-
**3**	0.22	8	0.3

**Table 3 micromachines-10-00294-t003:** Variance of equivalent input estimation under static condition with multi disturbances.

Methods	Variance (mV)	Reduction of Proposed Method
**KF**	0.212	1.9%
**LMS**	0.149	2.8%
**REM**	0.015	27.3%
**AEM**	0.004	-

**Table 4 micromachines-10-00294-t004:** Characteristic of multi disturbances in dynamic test.

Group	Amplitude (mm)	Frequency (Hz)	Duration (s)	Load Form
**1**	3	-	1.2 × 10^−5^	Pulse signal
**2**	1.7	77	4.5	Squared wave
**3**	2.2	18	1.6	Squared wave

**Table 5 micromachines-10-00294-t005:** Variance of the equivalent input under dynamic condition with multi disturbances.

Methods	Variance (mV)	Reduction of Proposed Method
**KF**	35.5	4.1%
**LMS**	2.07	70.1%
**REM**	2.0	72.5%
**AEM**	1.45	-

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
