# Peer review of "A Measurement-Data-Driven Control Approach towards Variance Reduction of Micromachined Resonant Accelerometer under Multi Unknown Disturbances"

_micromachines, 2019, doi:10.3390/mi10050294_

Round 1
Reviewer 1 Report
The authors present a control algorithm to reduce the influence of disturbances in MEMS sensors. They introduce the term "generalized MEMS sensors", however the manuscript seems to be focused on just one type of sensor: an accelerometer with resonant readout. In the introduction, reference is made to gyroscopes and accelerometers (ref. [1]-[4]), but it is not clear why these specific references were chosen. It would be better to refer to a text book or review paper. It is also not clear why ref. [6] was chosen, which presents a very specific inertial sensor.
Figure 1 is presented as schematic of a typical MEMS sensor. However, this is a very specific schematic that is certainly not applicable to a majority of MEMS sensors. The aythors should rewrite the manuscript and focus on the sensor that was actually used in the experiments. Furthermore, the experiments should be much clearer described: what signals were exactly provided to the vibration table and how does the presented algorithm really distinguish between an actual vibration signal and a disturbance? A reduction in variance does not seem a measure for the improvement in bias stability of the sensor. At least this needs further explanation.
Reviewer 2 Report
The manuscript describes a data-driven control approach for MEMS sensors to identify the multi-unknown disturbances. The manuscript composed well, nicely presented with the required data and sufficient scientific discussion. In my opinion, the work is suitable for publication in Micromachines journal, before that the authors need to clarify the below mentioned quires:
1. What is the response time of the proposed AEM method to identify the external/internal disturbances?
2. The likelihood function is dependent on the E, M steps. How can we estimate the accuracy of these steps, because every MEMS sensor (Linear and nonlinear data regions of the sensor) have a different set of unknown factors? Will the proposed AEM suitable for non-linear response based MEMS sensors or not?
Round 2
Reviewer 1 Report
The authors have addressed my previous remarks in adequately. Still, I think the paper can be further improved:
1) "Accuracy" is already a better term than "bias stability", but I think also an improved accuracy is not proven by the variance measurements. An improved accuracy would mean that the sensor output is a better measure for the applied acceleration, which in fact it is not (the "disturbance" is in fact an applied acceleration that should be detected). It would be better to talk about "improved performance" or to what is actually measured: "reduced variance".
I would also add "Resonant" to the title of the paper, and make it something like "A Measurement-Data-Driven Control Approach Towards Improving the Performance of a Micromachined Resonant Accelerometer Under Multi Unknown Disturbances"
2) I would add one or two references in the caption of figure 1 to examples of such accelerometers.
3) Section 4 could be improved by providing more information the resonant accelerometer that was used. Or add a reference to a publication that describes the accelerometer chip that was used. It would have been nice if a reference accelerometer had been included in the setup or if the displacement of the vibration table had been measured (e.g. optically). That would allow a comparison between the measurement and the actual applied acceleration and and a conclusion could be drawn with respect to the obtained accuracy. Also it would be interesting to see how the algorithm affects the dynamic response of the accelerometer, e.g. by doing a frequency sweep or by measurement of a step response.
4) I think the English grammer and spelling should be carefully checked throughout the paper. Some notable spelling mistakes: "microelectron mechanical" should be "microelectromechanical" and "acceleromter" should be "accelerometer".
Note to the editor: I cannot really judge the quality of the algorithm that was used since this is out of my field of expertise.
Author Response
Please check the attached file below.
